# Ninjin'yoeito reduces fatigue-like conditions by alleviating inflammation of the brain and skeletal muscles in aging mice

**Shotaro Otsuka** [1☉]*, **Ryoma Matsuzaki**[2☉], **Shogo Kakimoto**[2], **Yuta Tachibe**[2], **Takuya Kawatani**[2], **Seiya Takada**[3], **Akira Tani**[2], **Kazuki Nakanishi**[2], **Teruki Matsuoka**[2], **Yuki Kato**[2], **Masaki Inadome**[2], **Nao Nojima**[2], **Harutoshi Sakakima**[2], **Keita Mizuno**[4], **Yosuke Matsubara**[4], **Ikuro Maruyama**[5]*

1 Faculty of Welfare and Health Science, Oita University, Oita, Japan, 2 Faculty of Medicine, Department of Physical Therapy, School of Health Sciences, Kagoshima University, Kagoshima, Japan, 3 Department of Orthopedic Surgery, Graduate School of Medical and Dental Sciences, Kagoshima University, Kagoshima, Japan, 4 Kampo Research & Development Division, Tsumura Kampo Research Laboratories, Tsumura & Co., Ibaraki, Japan, 5 Department of Laboratory and Vascular Medicine, Graduate School of Medical and Dental Sciences, Kagoshima University, Kagoshima, Japan

☉ These authors contributed equally to this work.
* maruyama-i@eva.hi-ho.ne.jp (IM); k3360022@kadai.jp, otsuka-shotaro@oita-u.ac.jp (SO)

**Data Availability Statement:** All relevant data are within the manuscript and its Supporting Information files.

## Abstract

Fatigue can lead to several health issues and is particularly prevalent among elderly individuals with chronic inflammatory conditions. Ninjin'yoeito, a traditional Japanese herbal medicine, is used to address fatigue and malaise, anorexia, and anemia. This study aimed to examine whether relieving inflammation in the brain and skeletal muscle of senescence-accelerated mice prone 8 (SAMP8) could reduce fatigue-like conditions associated with aging. First, SAMP8 mice were divided into two groups, with and without ninjin'yoeito treatment. The ninjin'yoeito-treated group received a diet containing 3% ninjin'yoeito for a period of 4 months starting at 3 months of age. At 7 months of age, all mice underwent motor function, treadmill fatigue, and behavioral tests. They were then euthanized and the skeletal muscle weight, muscle cross-sectional area, and concentration of interleukin (IL)-1β and IL-1 receptor antagonist (IL-1RA) in both the brain and skeletal muscle were measured. The results showed that the ninjin'yoeito-treated group had higher motor function and spontaneous locomotor activity than the untreated group did and ran for significantly longer in the treadmill fatigue test. Moreover, larger muscle cross-sectional area, lower IL-1β concentrations, and higher IL-1RA concentrations were observed in both the brain and skeletal muscle tissues of the ninjin'yoeito-treated group than in the untreated group. The results suggest that ninjin'yoeito improves age-related inflammatory conditions in both the central and peripheral tissues and reduces fatigue.

## Introduction

Fatigue, which is a common condition among the elderly, increases the risk for various negative health outcomes, such as reduced physical capacity, increased susceptibility to disease, and

**Funding:** KM and YM received a research grant from Tsumura & Co. Tsumura & Co. had no role in the study design; collection, analysis, and interpretation of data; writing of the paper; and decision to submit for publication. Tsumura & Co. URL: https://www.tsumura.co.jp/

**Competing interests:** I have read the journal's policy and the authors of this manuscript have the following competing interests: KM and YM are company employees of the funding partner, Tsumura & Co. However, they have only funded the study and have no role in the study design, data collection and analysis, decision to publish, or preparation of the manuscript.

increased mortality [1,2]. Given that the global population is aging rapidly, the incidence of fatigue is on the rise, but no definitive interventions currently exist. Therefore, strategies to reduce fatigue are essential for promoting a long and healthy life.

Several animal studies using fatigue models have been conducted to improve fatigue; however, these studies have only reported models in which fatigue was induced by forced exercise or surgical procedures. Currently, animal models that show fatigue-like conditions owing to aging are lacking [3–6]. Therefore, it is crucial to establish a model that exhibits fatigue-like conditions caused by aging to discover an effective treatment for fatigue in the elderly.

Fatigue is divided into three main types: central (brain or mental) fatigue, peripheral (muscle) fatigue, and infection-induced fatigue [7]. Research has shown that inflammation-induced cytokines are an underlying mechanism for these types of fatigue [8,9]. Furthermore, chronic fatigue may be related to neuroinflammation caused by increased interleukin (IL)-1β levels in the brain [10]. In animal models of chronic fatigue syndrome, increased levels of the inflammation-inducing cytokine IL-1β have been observed in the prefrontal cortex region, which is considered the core of central fatigue [11,12]. In addition to inflammation, skeletal muscle age-related atrophy is another factor that contributes to chronic fatigue [9]. Thus, inhibiting IL-1β expression and preventing muscle atrophy in the brain and muscles may be important to reduce fatigue among aged patients with generalized inflammation and muscle atrophy.

The IL-1 receptor antagonist (IL-1RA) is an endogenous receptor antagonist in the cytokine signaling system that effectively and specifically blocks both IL-1α and β, thus inhibiting IL-1β [13]. Therefore, an increase in IL-1RA is expected to suppress chronic inflammation and reduce fatigue caused by IL-1β [10]. In animal studies, injections of IL-1RA into the brain have been shown to prolong the treadmill running time leading up to fatigue [14]. Furthermore, administering IL-1RA, a drug that inhibits IL-1 receptors, can result in decreased fatigue in patients with rheumatoid arthritis, a chronic inflammatory ailment [15]. Thus, the dynamics of IL1-related cytokines have been suggested to affect fatigue, and SAMP8, which has been reported to have a systemic inflammatory state and impaired physical function [16–18], may exhibit a fatigue-like conditions similar to that of elderly, but is not well understood. Furthermore, previous studies have administered drugs that inhibit IL-1 receptors, and it is difficult to use these drugs as an intervention method for fatigue reduction in countries where they are not approved by Pharmaceuticals and Medical Devices Agency (PMDA) in Japan. Therefore, the fatigue-reducing effects of dietary polyphenols and kampo medicines, which are expected to have anti-inflammatory effects, have attracted significant interest in recent years [19–21].

Ninjin'yoeito (NYT), a traditional Japanese kampo medicine, consists of 12 herbs: Rahmanian root, Japanese Angelica Root, Atractylodes Rhizome, Poria Sclerotium, Ginseng, Cinnamon Bark, Polygala Root, Peony Root, Citrus Unshiu Peel, Astragalus Root, Glycyrrhiza, and Schisandra Fruit, and is approved in Japan as a prescription drug to improve various disorders including declined constitution after recovery from disease, fatigue and malaise, anorexia, perspiration during sleep, cold limbs, and anemia [22]. To date, the ingredients in NYT have been shown to reduce fatigue caused by cancer and anticancer drugs [23,24]. Furthermore, several components in NYT have been reported to exert anti-inflammatory effects [25,26]. Specifically, orange peel extracts from citrus species, which contains polymethoxyflavones, have been found to increase IL-1RA expression in rat skeletal muscle tissue and cultured muscle cells [27]. Another study reported that polygala root reduces IL-1β expression and suppresses neuroinflammation in the brain of chronic restraint stress induced rats [28]. We have also previously shown that NYT inhibits age-related muscle atrophy in senescence-accelerated mice prone 8 (SAMP8), which exhibit characteristics similar to those of older humans, including shortened lifespan, anteversion of the spine, hair loss, decreased physical activity, and increased oxidative stress and inflammation [16–18]. Thus, NYT has the potential to reduce

age-related fatigue by suppressing the expression of IL-1β and increasing the expression of IL-1RA in the brain and skeletal muscle tissues and by suppressing muscle atrophy. However, to the best of our knowledge, studies that have revealed the effects of NYT on age-related fatigue are scarce.

Therefore, this study aimed to determine whether SAMP8, which exhibits characteristics similar to those of human aging, exhibits aging-induced fatigue-like conditions and to investigate the effects of NYT on the expression levels of IL-1β and IL-1RA in the brain and skeletal muscle to determine whether NYT promotes a reduction in aging-induced fatigue-like conditions.

## Materials and methods

### Animals

Sixteen 7-week-old male SAMP8 (weight: 26.5 ± 2.0 g) and sixteen age-matched male SAM resistant1 (SAMR1) (29.7 ± 1.3 g) mice as the normal subject group were purchased from SLC, Japan (Hamamatsu, Japan). Two to three mice per cage were maintained on a 12-h light/dark cycle with temperature controlled at 23.0 ± 1.0°C. Researchers involved in all animal experiments in this study received customized training (in animal care and handling) designed by the Institute of Laboratory Animal Sciences of Kagoshima University, prior to performing any experiments. The experimental protocol was approved by the Ethics Board of the Institute of Laboratory Animal Sciences of Kagoshima University (approval number: MD18056).

### NYT preparation

At three months of age, sixteen SAMP8 mice were randomly divided into two groups: one with NYT treatment (SAMP8 NYT, n = 8) and the other without NYT treatment (SAMP8 Control, n = 8). Sixteen SAMR1 mice were randomly divided into two groups: one with NYT treatment (n = 8) and another without NYT treatment (n = 8).

NYT was administered as described in our previous study [18]. Specifically, NYT (ninjin'yoeito TJ-108, lot no. 362113100) was obtained from Tsumura & Co. The NYT comprised 12 types of herbs, including 4.0 g of Rahmanian root, 4.0 g of Japanese Angelica Root, 4.0 g of Atractylodes Rhizome, 4.0 g of Poria Sclerotium, 3.0 g of Ginseng, 2.5 g of Cinnamon Bark, 2.0 g of Polygala Root, 2.0 g of Peony Root, 2.0 g of Citrus Unshiu Peel, 1.5 g of Astragalus Root, 1.0 g of Glycyrrhiza, and 1.0 g of Schisandra Fruit. The botanical names were verified using World Flora Online (www.worldfloraonline.org) and MPNS (http://mpns.kew.org). The quality of the NYT ingredients was standardized based on Good Manufacturing Practices established by the Japanese Ministry of Health, Labor, and Welfare. NYT was approved by the PMDA of Japan (approval number: 16100AMZ03305000). The detailed botanical origin and medicinal properties of NYT have been described in a study by Matsumoto et al. [29]. Extensive profiling of the NYT components was performed by Tsumura & Co. using three-dimensional HPLC (S1 Fig). NYT was mixed with standard feed (Oriental Yeast Co., Ltd., Tokyo, Japan) at a concentration of 3% (w/w) and fed to the NYT-treated groups from three to seven months of age for 17 weeks. The no NYT treatment group was fed a standard diet (KBTO190189; Oriental Yeast Co., Tokyo, Japan). The amount of feed given to each mouse was 5 g/day (amount of feed given per cage: 5 g × 2 or 3 mice × 7 days = 70 or 105 g), and food and water were allowed for free consumption by the mice. Feed intake was measured once per week and replaced with fresh feed. The average weekly feed intake and NYT dose were then calculated (S2 Fig). The NYT dose was calculated as feed intake (g) × 0.03. The mean daily doses of NYT per mouse during the intervention period were SAMP8:0.121 ± 0.002 g and SAMR1:0.121 ± 0.002 g (mean ± standard error).

The intervention period lasted from 3 to 7 months of age; food intake and body weight were measured periodically during the NYT intervention period to assess health status. The humane endpoints that warranted euthanasia were significant weight loss (>20% weight loss from peak body weight), failure to ingest food, and decreased activity in the cage. If a mouse met one or more of the endpoint criteria, euthanasia was required within 2 h. These endpoints were not reached during the experimental period. Additionally, none of the individuals died from either SAMR1 or SAMP8 before the age of 7 months.

## Motor function test

We assessed locomotor function in all mice using rotarods, as described previously [18]. Specifically, mice were placed on a rotarod cylinder and the time it took for them to fall out of the cylinder was measured. Rotation speed was increased from 0 rpm to 40 rpm in 4.0 rpm increments at 60 second intervals; the cut off value was 100 s. The test was performed twice, and the average time (in seconds) was used for analysis. Referring to previous studies [16–18], the tests were conducted between 10:00 a.m. and 1:00 p.m.

## Muscle strength grip-strength test

The grip strength of all mice was measured using a method described in a previous study [18]. Specifically, using a small animal grip strength measuring device (MK-380si; Muromachi Kikai Co., Ltd., Tokyo, Japan), mice were made to grasp a wire mesh and their tails were pulled gradually until they released their hands. Maximum force (kg) was measured twice, and the average value was calculated. Referring to previous studies [16–18], the test was conducted between 10:00 a.m. and 1:00 p.m.

## Behavioral test

Activity was assessed using an open field test as described previously [18]. Specifically, the open field apparatus (55 cm × 60 cm × 40 cm) was placed in a quiet environment and wiped clean with a 70% ethanol solution between trials. Mice were habituated in the center of the open field for 5 min and filmed for 1 h with a video camera (Logitech HD Pro Webcam C920r; Logitech Co., Ltd., Lausanne, Switzerland) mounted above the open field. The SMART Ver3.0 video camera system (Panlab, Barcelona, Spain) was used to measure the time that mice were motionless in the open field; the time of stillness was compared to the time of stillness at 6 months of age and the percentage increase in stillness at 7 months of age was calculated. Referring to previous studies [16–18], tests were conducted between 10:00 a.m. and 5:00 p.m. The brightness of the room lighting during the tests ranged from 150 to 300 lux.

## Treadmill fatigue test

A modified protocol for fatigue testing on a treadmill was used [30]. Specifically, one mouse was placed in each lane of the treadmill and started at a speed of 12 m/min. Thirty seconds after the start, the speed was increased to 14 m/min, followed by increasing to 16 m/min and 18 m/min at 1 min and 6 min later, respectively; the speed was not changed further from 18 m/min. If the runner remained on the wire mesh behind the treadmill for 5 s within the first 30 min of the start, they were considered fatigued and the time they ran was recorded. Referring to previous studies [16–18], the test was conducted between 10:00 a.m. and 5:00 p.m.

## Euthanasia

Sixteen SAMP8 and sixteen SAMR1 mice were used in this study. All mice completed all tests at 7 months of age, the end of the intervention. Two days after the treadmill fatigue test at 7 months of age, all mice were anesthetized with an intraperitoneal injection of sodium pentobarbital (100 mg/kg), blood was collected from a vein in the abdomen, followed by de-bleeding with heparinized saline. The animals were then decapitated, and the prefrontal cortex and brain cortex, as well as the right and left gastrocnemius muscles, were sampled. The gastrocnemius muscles were measured for muscle wet weight after collection.

## Tissue preparation

The prefrontal cortex was fixed overnight at 4°C in 4% paraformaldehyde in 0.1 M phosphate buffer (pH 7.4). After fixation, the tissues were dehydrated, and embedded in paraffin. The paraffin-embedded brain tissue was prepared as 4 μm-thick coronal sections. The coronal section was obtained 1 mm cranial from the bregma.

The gastrocnemius muscle taken from the right hindlimb was mounted vertically on a cork plate in tragacanth rubber jelly to obtain a central section of the muscle. This was quickly frozen in isopentane chilled with liquid nitrogen and stored at -80°C for subsequent analysis. A cryostat microtome was used to prepare 10 μm transverse sections at -20°C. The prepared brain and muscle sections were used for immunohistochemical staining. The left gastrocnemius muscle and brain cortex were stored at -80°C until used for western blotting.

## Analysis of blood biomarkers

The collected blood was placed in a blood collection tube (TERUMO Co., Tokyo, Japan) containing a serum separator and centrifuged at $3000 \times g$ for 10 min to obtain serum. Corticosterone analysis was performed by Yanaihara Institute Inc. (Shizuoka, Japan) using the obtained serum. Other fatigue markers in blood were analyzed at the Nagahama Plant of Oriental Fermentation Industry Co. (Shiga, Japan). Serum was stored at -80°C until analysis.

## Histology and immunohistochemistry

Brain and gastrocnemius sections were stained with the following antibodies: rabbit monoclonal anti-Iba-1 (marker of microglia) (ab178846; Abcam, Cambridge, UK), mouse monoclonal anti-skeletal myosin (fast) (marker for skeletal muscle Type II fibers) (M4276; Sigma-Aldrich, Saint Louis, MO, USA), rabbit monoclonal anti-slow skeletal myosin heavy chain (marker for skeletal muscle Type I fibers) (ab234431; Abcam), and rat monoclonal anti F4/80 (marker for macrophages) (ab6640; Abcam).

Brain sections were deparaffinized and endogenous peroxidase activity was inhibited by treatment with 3% methanol peroxide for 10 min. The sections were then washed three times (5 min each) in phosphate-buffered saline (PBS, pH 7.6) and blocked with 10% skim milk-PBS for 20 min. Brain sections were then reacted with rabbit anti-Iba-1 antibody (1:1000) overnight. The next day, sections were washed three times (5 min each) with PBS, reacted with peroxidase-labeled dextran polymer (EnVision; Dako, CA, USA) for 60 min at room temperature (24–25°C), and immunoreactivity was observed by diaminobenzidine staining.

Muscle sections were examined by immunofluorescence staining for rat antiF4/80 antibody (1:1000) and 4′,6-diamido-2-phenylindole staining. In addition, mouse anti-skeletal myosin (fast) antibody (1:200) and rabbit anti-slow skeletal myosin heavy chain (1:200) were used to stain each type of muscle fiber. After immersion in acetone at 4°C for 10 min and subsequent fixation, endogenous peroxidase activity was inhibited and blocked in the same way as for the

brain tissue, and then reacted with the primary antibodies described above overnight. The next day, after washing three times (5 min each) with PBS, sections were treated for 60 min with Alexa Fluor 488-labeled goat anti-rat IgG antibody (1:200, ab150157; Abcam), Alexa Fluor 488-labeled goat anti-rabbit IgG antibody (1:200, ab150077; Abcam), and Alexa Fluor 555-labeled goat anti-mouse IgG antibody (1:200, ab150114; Abcam). The cells were then washed three times (5 min each) with PBS, and sections stained with F4/80 were counter-stained with 4′,6-diamido-2-phenylindole (1:500, 340–07971; Dojindo Laboratories, Kumamoto, Japan) for 10 min. Finally, sections were mounted in aqueous mounting solution and observed under a fluorescence microscope (EVOS M5000 Imaging system; AMG, Mill Creek, USA).

## Quantitative analysis of immunostained sections

Brain sections stained with Iba-1 were photographed at two locations each in the cortex and striatum at 40× magnification to avoid overlap per mouse, and the number of Iba-1-positive cells was counted (S3A Fig). The number of positive cells from four images was averaged to determine the number of positive cells per image stained with F4/80. Gastrocnemius muscle sections were photographed in two locations at 10× magnification to avoid overlap per mouse, and the number of F4/80-positive cells and nuclei merging was determined based on a previously described method (S3B Fig) [31]. Gastrocnemius muscle sections stained for each muscle fiber type were photographed in one deep section at 10× magnification, 20–50 fibers for Type 1 fibers and 100 fibers for Type 2 fibers were randomly selected, and the cross-sectional area was measured and averaged per individual. All quantitative analyses were performed using Image J version 1.46r (NIH, USA) and were performed by two individuals who were blinded to each group.

## Western blotting

Western blotting was performed to detect protein levels in the cortex and gastrocnemius muscle of 7-month-old SAMP8 and SAMR1 (n = 4 per group). Each tissue was placed on ice and homogenized with T-Per reagent (78510; Pierce Biotechnology, Rockford, USA). Approximately 10 μg protein per sample was loaded onto 4–20% mini-protean precast gel (Bio-Rad, Hercules, CA, USA) and transferred to polyvinylidene fluoride or nitrocellulose membranes. The membranes were blocked with Tris-buffered saline containing 5% skim milk + Tween 20 for 1 h at room temperature and then reacted with the primary antibody overnight at 4˚C. The next day, the antibody was reacted with goat anti-rabbit IgG H&L antibody (1:2000, ab97051; Abcam) for 1 h at room temperature. The following primary antibodies were used for western blotting: rabbit polyclonal anti IL-1β (1:1000, ab9722; Abcam), rabbit monoclonal anti-IL-1RA (1:1000, ab124962; Abcam), rabbit monoclonal anti-Iba-1 (1:1000, ab178846; Abcam), mouse monoclonal anti-α-tubulin (1:2000, 66031-1-lg; Proteintech Group, Inc., USA), and mouse monoclonal anti-GAPDH (1:10000, #AM4300; Invitrogen, USA). Protein bands were visualized with a chemiluminescence device (WSE-6100 Lumino Graph I, ATTO, Tokyo, Japan) and measured using Image J version 1.46r (NIH, USA). The intensity of the target band was normalized to that of the α-tubulin or GAPDH band of the same sample.

## Statistical analysis

Statistical analysis was performed using the Shapiro–Wilk test to confirm normality, followed by parametric or nonparametric tests. Data were analyzed using either one-way analysis of variance (ANOVA) or the Kruskal–Wallis test, and Tukey's post-hoc test was applied for multiple comparisons. Statistical significance was set at $p < 0.05$. Data are presented as

mean ± standard error. All data were analyzed using IBM SPSS Statistics, version 26 (IBM Corp., Armonk, NY, USA).

## Results

### NYT does not affect the physical function of SAMR1 mice

To determine whether NYT alters the physical function of SAMR1 mice with normal aging, SAMR1 mice treated with and without NYT were compared. Body weight, walking time in the rotarod test, grip strength, treadmill fatigue test, gastrocnemius weight, and cross-sectional area did not show significant changes after NYT treatment (Fig 1A, 1C–1E and 1H-1K). Ninjin'yoeito administration increased food intake and blood corticosterone levels and decreased resting time in the open field test; however, the differences were not statistically significant (Fig 1B, 1F and 1G). These results suggest that NYT does not enhance the physical function of SAMR1.

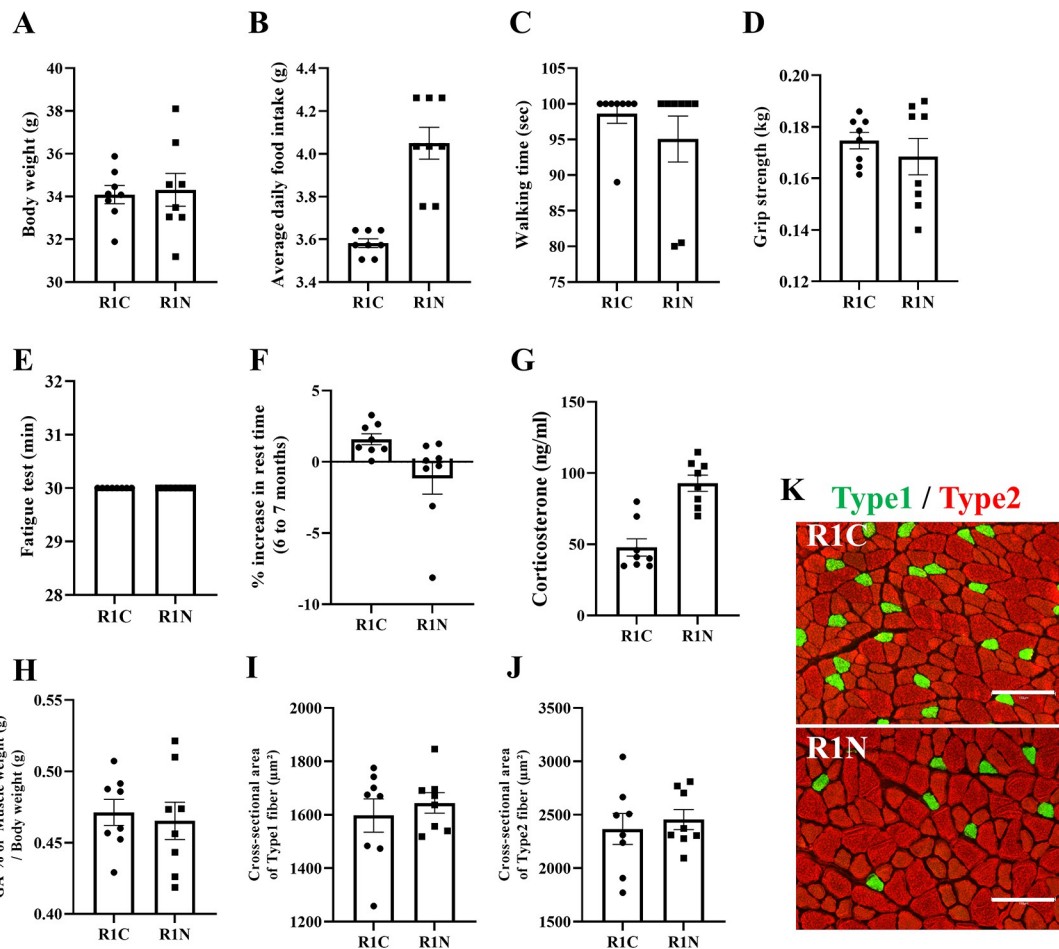

**Fig 1. Effects of NYT on body weight, appetite, physical function, fatigue, and gastrocnemius muscle in SAMR1.** Body weight at 7 months of age (A), average daily food intake (B), rotarod test results (C), grip strength results (D), treadmill fatigue test results (E), percentage increase in resting time in the open field test (F), blood corticosterone concentration (G), and gastrocnemius weight-to-weight ratio (H). Cross-sectional area of Type 1 fibers (I), cross-sectional area of Type 2 fibers (J), and representative fluorescent stained images (green: Type 1 fibers, red: Type 2)(K). Feeding NYT to SAMR1 mice had no significant effects on body weight, food intake, physical function, fatigue, or the gastrocnemius muscle. "R1C," SAMR1 mice fed the Control diet (n = 8); "R1N," SAMR1 mice fed NYT (n = 8). Data are shown as mean ± standard error. Scale bar = 150 μm.

## NYT improves aging-related symptoms

The effects of NYT on aging-related weight loss, anorexia, and loss of locomotor function were examined. Body weight at 6 and 7 months of age did not differ among the three groups (Fig 2A); the rate of weight loss at 7 months of age was significantly higher in the two SAMP8 groups than in the SAMR1 group which exhibited a peak weight at 6 months of age ($p < 0.01$). However, the rate of weight loss was significantly lower in the SAMP8 NYT group than in the SAMP8 Control group (Fig 2B, $p < 0.05$). Furthermore, the SAMP8 NYT group had the highest average daily food intake (Fig 2C). In the rotarod test, which evaluates motor function, the SAMP8 Control group had a significantly shorter walking time than did the SAMR1 group (Fig 2D, $p < 0.01$). Similarly, the SAMP8 Control group was significantly weaker than the SAMR1 group in grip strength (Fig 2E, $p < 0.01$). However, the SAMP8 NYT group showed significant improvement over the SAMP8 Control group in walking time and grip strength in the rotarod test (Fig 2D and 2E, $p < 0.01$). These results suggest that NYT administration improves weight loss, anorexia, and motor function associated with aging.

## NYT reduces fatigue-like conditions associated with aging

We investigated whether NYT administration can reduce the fatigue-like conditions associated with aging. In the treadmill fatigue test, compared to SAMR1, the SAMP8 Control group had a

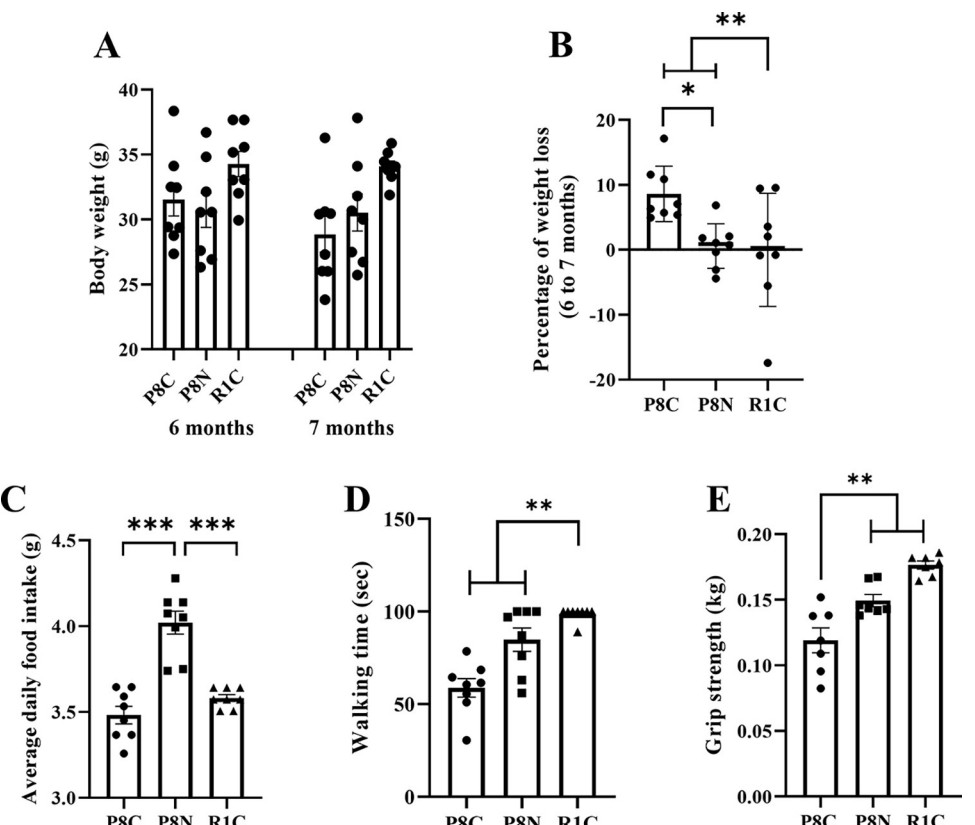

**Fig 2. Effect of NYT on age-related weight loss, anorexia, and motor function decline.** Weight at 6 and 7 months of age (A), percentage weight loss comparing mice at 6 and 7 months of age (B), average daily food intake (C), rotarod test results (D), and grip strength results (E). Aging significantly reduced body weight, motor function and grip strength; weight loss, anorexia, and motor function decline were significantly improved in the SAMP8 NYT group ("P8N," n = 8) compared to those in the SAMP8 Control group ("P8C," n = 8) (B-E). "R1C," normal control SAMR1 (n = 8). Data are presented as mean ± standard error. *p<0.05, **p<0.01, ***p<0.001.

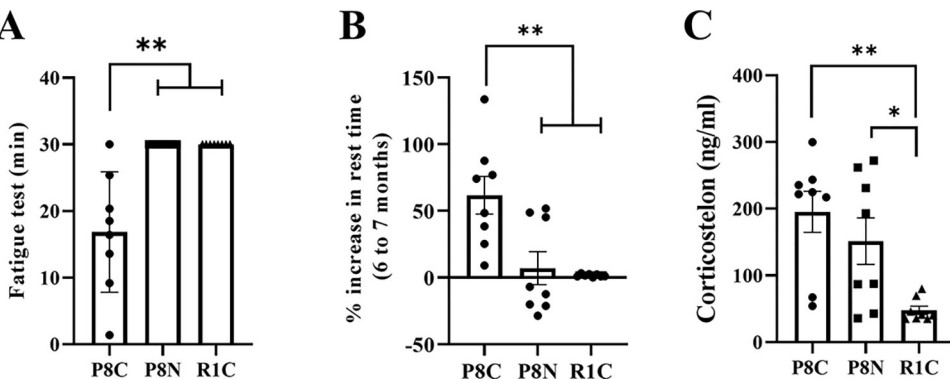

**Fig 3. Effect of NYT on age-related fatigue-like conditions.** The treadmill fatigue test (A), percentage increase in resting time in the open field test (B), and blood corticosterone concentration (C). The SAMP8 Control group ("P8C," n = 8) exhibited a significantly faster time to fatigue and significantly increased time immobile in the open field test than that in the normal control, SAMR1 ("R1C," n = 8). NYT intake significantly improved the time to fatigue and increased stationary time. Corticosterone was increased in both SAMP8 groups. Data are presented as mean ± standard error. *p<0.05, **p<0.01.

short running time, but NYT administration significantly increased running time (Fig 3A, p < 0.01). The SAMP8 Control group showed a significant increase in resting time over the SAMR1 group (Fig 3B, p < 0.01), but NYT administration prevented the increase in resting time (Fig 3B). Blood corticosterone levels, a marker of fatigue, were significantly higher in the two SAMP8 groups than in SAMR1 (Fig 3C, p < 0.01), with no significant changes owing to NYT administration (Fig 3C). Furthermore, only blood urea nitrogen showed an age-related increase, whereas the other blood fatigue markers did not show significant changes (S4 Fig). These results suggest that SAMP8 exhibits fatigue-like conditions associated with aging and that NYT administration improves fatigue-like conditions.

## NYT improves muscle cross-sectional atrophy associated with aging

We examined the effect of NYT on aging skeletal muscles. The SAMP8 Control group had significantly lower muscle weight than the SAMR1 group did (Fig 4A, p < 0.01). Furthermore, considering cross-sectional area, both Type 1 and Type 2 fibers were significantly smaller in the SAMP8 Control group than in the SAMR1 group (Fig 4B–4D, p < 0.01). However, NYT treatment did not change Type 1 fibers (Fig 4B and 4D). In contrast, muscle weight loss was reduced (Fig 4A) and the reduction in cross-sectional area of type 2 fibers was significantly inhibited (Fig 4C and 4D, p < 0.05). These results suggest that NYT improves the atrophy of muscle cross-sectional area associated with aging in the gastrocnemius muscle.

## NYT suppresses neuroinflammation associated with aging in the brain

The effects of NYT on microglial activation and cytokines in the brain were investigated. The number of Iba-1-positive cells in the prefrontal cortex did not differ among the three groups, SAMR1, SAMP8 Control, and SAMP8 NYT (Fig 5A and 5B). Western blotting analysis showed that the SAMP8 Control group exhibited significantly increased expression of the proinflammatory cytokine IL-1β (Fig 5C and 5D, p < 0.01) and significantly decreased expression of the anti-inflammatory IL-1RA compared to expression levels in the SAMR1 group (Fig 5C and 5E, p < 0.01). However, NYT treatment significantly suppressed the age-related increase in IL-1β expression (Fig 5C and 5D, p < 0.05) and decrease in IL-1RA expression (Fig 5C and 5E, p < 0.05). The levels of Iba-1 expression did not differ among the three groups, as

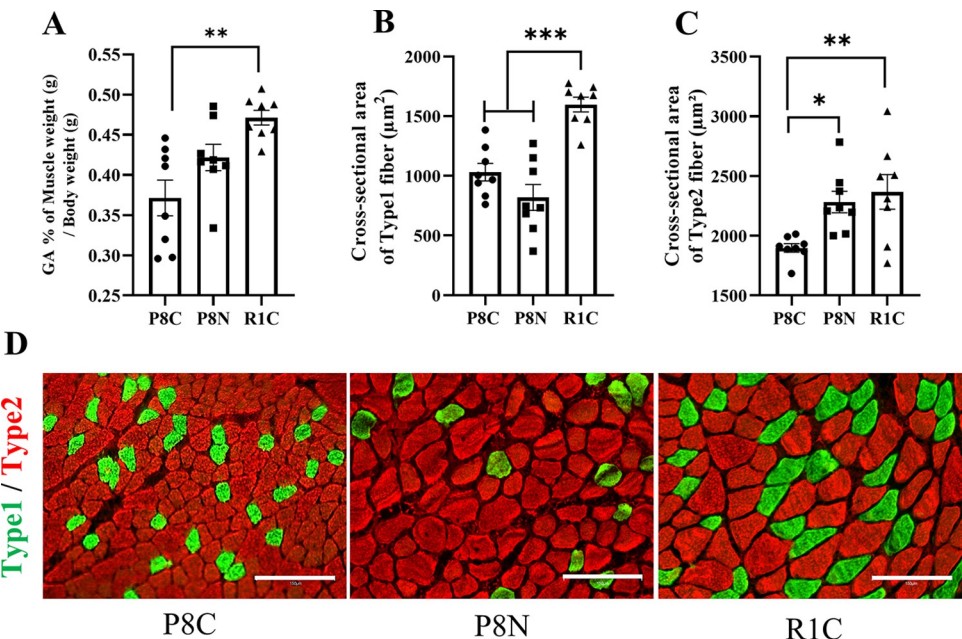

**Fig 4. Effect of NYT on weight and cross-sectional area of aged gastrocnemius muscle.** Gastrocnemius muscle weight-to-body weight ratio (A), cross-sectional area of Type 1 fibers (B), cross-sectional area of Type 2 fibers (C), and representative fluorescent stained images (green: Type 1 fibers, red: Type 2) (D). The SAMP8 Control group ("P8C," n = 8) exhibited significantly reduced muscle weight and cross-sectional area compared to that in the normal control SAMR1 ("R1C," n = 8) The SAMP8 NYT group ("P8N," n = 8) exhibited heavier muscle weight and significantly larger Type 2 fibers than did the SAMP8 Control group. Data are presented as mean ± standard error. *p<0.05, **p<0.01, ***p<0.001; scale bar = 150 μm.

also immunoreactivity (Fig 5C and 5F). These results suggest that NYT administration suppresses aging-induced inflammation in the brain, a central tissue.

## NYT improves aging-dependent inflammatory2 state in the gastrocnemius muscle

We investigated whether NYT improves the age-related inflammatory state in the skeletal muscle as it does in the brain tissue. The expression of cytokine-expressing macrophages in the skeletal muscle was evaluated using F4/80 nuclear staining and the number of positive cell nuclei was significantly increased in the SAMP8 Control group compared to that in the SAMR1 group (Fig 6A and 6B, p < 0.001). Furthermore, western blotting analysis showed increased expression of IL-1β (Fig 6C and 6D, p < 0.01) and decreased expression of IL-1RA (Fig 6C and 6E, p < 0.01). NYT administration significantly improved the increase in macrophages (Fig 6A and 6B, p < 0.05), increased expression of IL-1β (Fig 6C and 6D, p <0.05), and decreased expression of IL-1RA (Fig 6C and 6E, p < 0.05). These results suggest that NYT administration suppresses aging-induced inflammation even in the skeletal muscle, a peripheral tissue.

## Discussion

In this study, we investigated whether SAMP8 causes fatigue-like conditions with aging. We also investigated whether NYT administration can inhibit age-related motor function decline and reduce fatigue-like conditions, and whether NYT administration can inhibit age-related brain and skeletal muscle inflammation and muscle atrophy. The results showed that

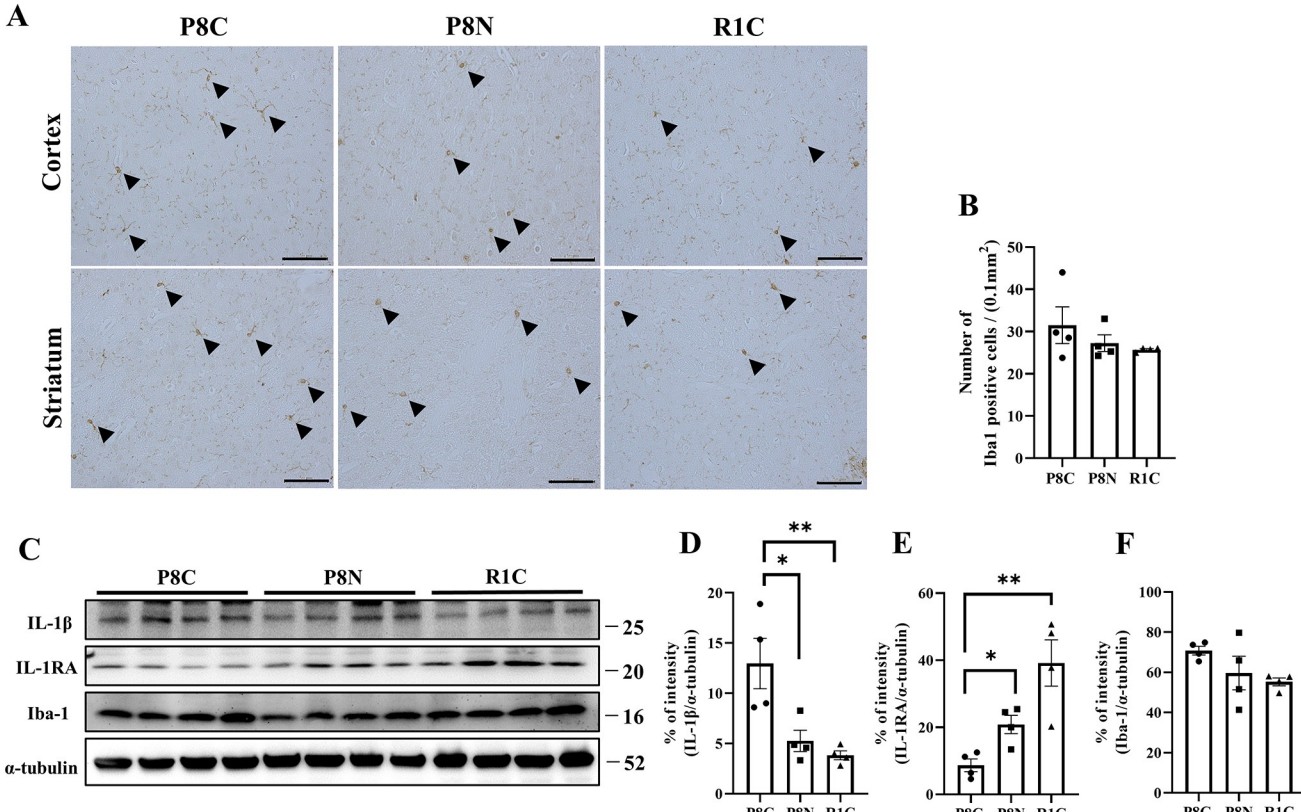

**Fig 5. Effect of NYT on age-related brain inflammation.** Representative images of Iba-1 immunoreactivity in brain tissue (A). The number of Iba-1 positive cells did not change significantly (B). Representative western blots and semi-quantitative analyses of the expressions of IL-1β, IL-1RA, and Iba-1 (C). IL-1β expression was significantly lower in the SAMP8 NYT group ("P8N") than in the SAMP8 Control group ("P8C") (D). IL-1RA expression was significantly higher in the SAMP8 NYT group than in the SAMP8 Control group (E). "R1C," normal control SAMR1. No significant differences in the expression of Iba-1 protein were observed (F). Data are shown as mean ± standard error. *p<0.05, **p<0.01; scale bar = 50 μm (A), n = 4 per group for all data (B, D, E, F).

7-month-old SAMP8 exhibited fatigue-like conditions, and NYT administration suppressed weight loss, anorexia, and motor function decline, and prolonged treadmill running time to fatigue, but did not increase resting time in the open field test. In addition, NYT treatment also showed an improvement in inflammatory status and prevention of muscle cross-sectional area atrophy by suppressing IL-1β expression and increasing IL-1RA expression in the brain and skeletal muscle tissues. These results suggest that NYT may prevent age-related decline in motor function as well as improve central and peripheral inflammation and reduce fatigue in the elderly.

Several animal studies have been conducted to develop treatments to reduce fatigue; however, these investigations used artificially induced fatigue models [3,5,11,12]. Therefore, no effective therapeutic strategy for aging-induced fatigue observed in the elderly has been found. Therefore, we hypothesized that SAMP8, which exhibits characteristics similar to those of human aging and was used in our previous study, would exhibit aging-induced fatigue-like conditions. Seven-month-old SAMP8 exhibited characteristics of aging such as weight loss and decreased motor function, as well as symptoms of fatigue such as decreased activity. Furthermore, several blood biomarkers, which are known to increase with fatigue, were increased in SAMP8, supporting the idea that they were fatigued [32,33]. Interestingly, there was an increase in IL-1β and a decrease in IL-1RA, both of which are believed to play an important

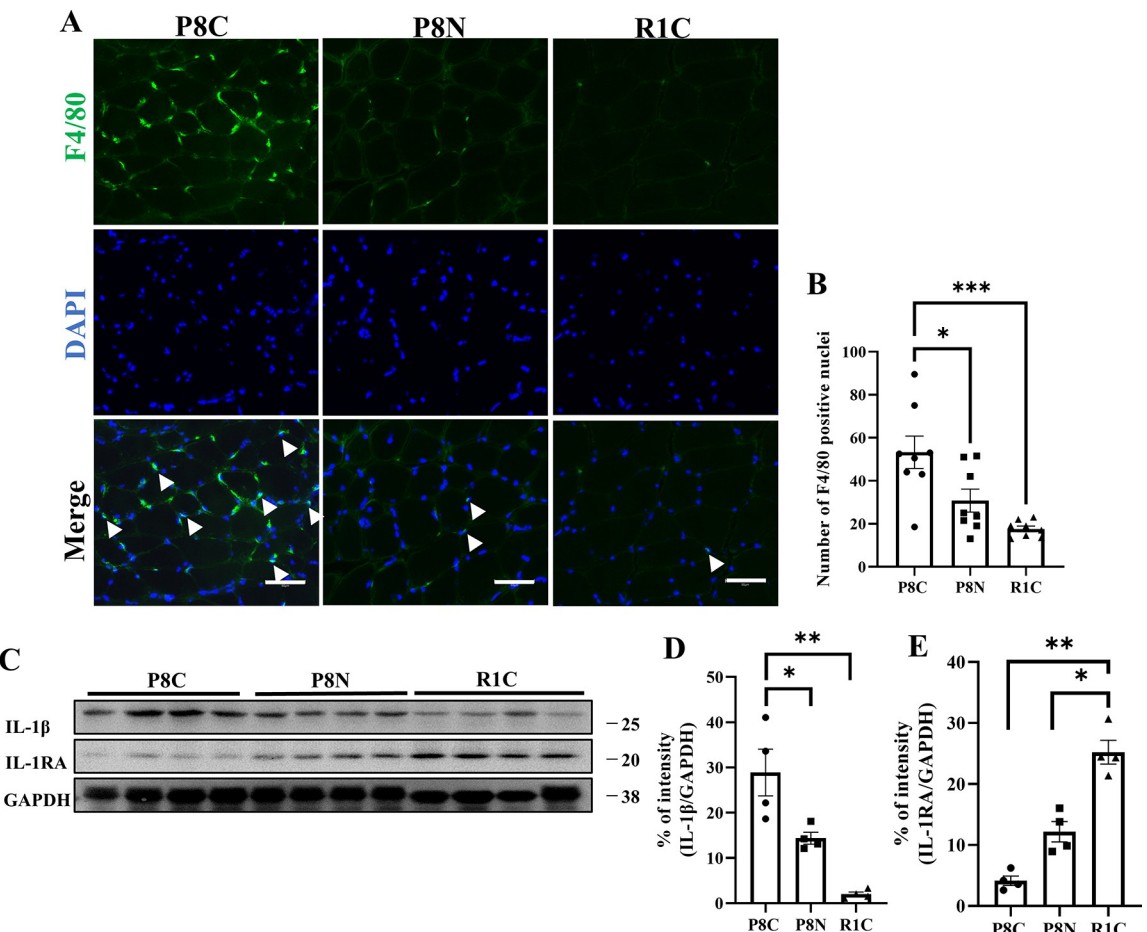

**Fig 6. Effect of NYT on inflammation in aged gastrocnemius muscle.** Co-stained images of F4/80 and DAPI in gastrocnemius muscle (A), The SAMP8 Control group ("P8C") showed increased macrophages (B) Representative western blots and semi-quantitative analyses of the expressions of IL-1β and IL-1RA (C). The SAMP8 Control group ("P8C") showed increased expression of IL-1β (D), and decreased expression of IL-1RA (E) compared to those in the normal control SAMR1 ("R1C"). Inflammation was significantly suppressed in the SAMP8 NYT group ("P8N") compared to that in the SAMP8 Control group. Data are shown as mean ± standard error. *p<0.05, **p<0.01, ***p<0.001; scale bar = 50 μm (A). (A) and (C) were analyzed with n = 8 and n = 4 per group, respectively.

role in fatigue, in both the brain and muscle of aged SAMP8. These findings indicate that SAMP8 exhibits characteristics of aging and fatigue-like conditions similar to those of humans. Therefore, SAMP8 could be used as a new animal model for exhibiting fatigue-like conditions with aging.

Inflammatory or anti-inflammatory cytokines and muscle atrophy have been reported to be involved in the major mechanisms of fatigue [8,9]. Cytokine-focused studies of fatigue have investigated TNF-α, IL-6, and IL-10 [34]. Among the various cytokines, the inflammatory cytokine IL-1β and IL-1RA, the anti-inflammatory action of which is the blocking of its receptor, have been reported to play an important role in fatigue [14,35]. Several components in NYT are known to act against these cytokines. For example, ligustilide, the main active ingredient of Toukai, and 1-deoxynojirimycin, a major alkaloid component, have been reported to downregulate IL-1β in the brain of SAMP8 and reduce neuroinflammation [36,37]. In previous studies, nobiletin, a polymethoxyflavones that increase IL-1RA expression in muscles, also appeared to cross the blood-brain barrier [27,29]. Thus, it is possible that components of NYT may directly increase IL-1RA expression in the brain. In addition, microglia are the major

mediators of innate immune responses in brain tissue, and several components in NYT have been reported to inhibit microglial activation [38–40]. Therefore, we expected that NYT would inhibit microglial activation and ameliorate neuroinflammation. Noteworthily, however, in the present study, we found significant changes in cytokines even though NYT treatment did not alter microglial activation; ginsenosides, the active ingredient of ginseng in NYT, appear to induce M2 macrophage polarization [41]. We hypothesize that although the effect of NYT on microglial polarization is not known, NYT administration may have induced M2 microglial polarization and alleviated inflammation.

The anti-inflammatory effects of NYT components have also been reported in the skeletal muscle [27], and in the present study, administration of NYT inhibited IL-1β and increased IL-1RA in the skeletal muscle. Aging appears to cause an increase in inflammatory cytokines and macrophage infiltration in the skeletal muscle [42,43]. In this study, the number of macrophages was investigated by fluorescent staining and was significantly higher in the aged SAMP8 gastrocnemius muscle. This confirms that NYT administration alleviated inflammation, as evidenced by the low number of macrophages in the gastrocnemius muscle of NYT-treated SAMP8. In addition to inflammation, fatigue resistance is reduced in the skeletal muscle owing to atrophy of muscle cross-sectional area [44]. Physical activity and high food intake are important for maintaining skeletal muscle cross-sectional area [45]. The effects of NYT on anorexia are well known, with increased food intake reported by activating orexin-1 receptors in the hypothalamus [46]. Therefore, the administration of NYT may have reduced the decrease in food intake owing to aging, and the maintenance of muscle cross-sectional area may also have contributed to the reduction in fatigue.

In the present study, even though NYT administration was found to reduce fatigue-like conditions, it had no effect on blood biomarkers (S4 Fig). Blood biomarkers are also important indicators, and increased biomarkers are associated with fatigue [32,33,47]. However, the results of the present study were inconsistent with those of previous studies. The blood biomarkers used in this study have been reported to increase during or immediately after exercise [4]. In the present study, blood samples were taken 2 days after the treadmill fatigue test was performed, which may not have reflected the results. In summary, our findings suggest that NYT may reduce aging-related fatigue by alleviating inflammation in both the central tissue (brain) and the peripheral tissue (skeletal muscle) and preventing muscle cross-sectional area atrophy.

In this study, NYT was shown to prevent decline in physical function and reduce fatigue owing to aging, but whether it improved function beforehand or attenuated the rate of functional decline remains unclear. Therefore, we administered NYT to normal mice, SAMR1, and compared them to the untreated group. Administration of NYT to normal mice did not improve physical function or cause muscle hypertrophy. Therefore, NYT may not improve physical function, but only attenuate the rate of age-related decline in physical function.

Our study has several limitations. First, the method of NYT administration in this study was the same as that used in previous studies [18,48]; mice were fed NYT mixed with standard diets. Although NYT intake differed little among the individuals and did not change significantly during the intervention period (S2 Fig), the NYT dose for each mouse was not standardized. Therefore, the results of this study suggest that NYT may reduce age-related fatigue. However, further research is needed to determine the detailed dosage of NYT that is effective for clinical applications. Second, this study investigated the effects of NYT on SAMR1 in normal mice. These results confirmed that NYT administration did not alter SAMR1. Therefore, a comparison was made among the three groups: SAMR1 fed a normal diet was used as the normal control group, and SAMP8 was used as the group that was administered NYT or not. Third, we used only SAMP8 in this study. Reports on models of fatigue-like conditions owing

to aging are lacking, and we used SAMP8 in the present study because we believed that SAMP8, which was used in our previous study, may present fatigue-like conditions [18]. In addition to SAMP8, other models have been used in aging studies, including old C57BL/6J mice. For clinical application, NYT needs to be studied in other aged animals to accumulate evidence that it reduces fatigue owing to aging. Forth, blood biomarkers were used in this study because they have been used as an indicator of fatigue; however, blood samples were collected 2 days after the treadmill fatigue test rather than immediately after. As blood biomarkers increase during or immediately after exercise, they may not reflect the effect of NYT, and whether NYT affected blood biomarkers remains unclear. Furthermore, this study revealed that IL-1-related cytokines are altered in the tissues of SAMP8 mice exhibiting fatigue-like conditions, and future studies are needed to determine whether these cytokines can be used as blood biomarkers for age-related fatigue. Fifth, in this study, the skeletal muscle was analyzed only in the gastrocnemius muscle. The gastrocnemius muscle is the most important muscle for hindlimb activity in mice and the most affected by aging [49]. Therefore, it is the most used muscle in studies of aging skeletal muscles; we also used the gastrocnemius muscle in our current study. However, future studies should investigate whether NYT affects other muscles as well. Additionally, the administration of NYT prevented a reduction in the cross-sectional area of Type 2 fibers but did not affect the cross-sectional area of Type 1 fibers. Therefore, further investigations are required to determine the effect of NYT on each muscle fiber type in aging muscles. Finally, it appears that regulation of systemic cytokine expression levels is important in reducing fatigue [6]. Therefore, the present study focused on cytokine expression levels in the brain and muscle and did not examine whether NYT acted on microglial polarization. Further studies are needed to investigate the possibility that microglial polarization may underlie the differences in cytokine expression after NYT administration in the brain. This study investigated inflammation in the brain using the prefrontal cortex, including both the cortex and striatum, using western blotting. Immunohistochemical analysis was also conducted on the cortex and striatum of the prefrontal cortex. The study did not focus on the specific functions of the striatum but rather assessed microglia in the prefrontal cortex. Future research should aim to investigate striatal regions and motivation. Despite these limitations, NYT administration not only improved age-related weight loss, anorexia, decreased motor function, and decreased grip strength, but also reduced fatigue by alleviating inflammation in both the central brain and peripheral skeletal muscle tissues. In our previous study, we found that SAMP8 used in the current experiment exhibited physical frailty-like symptoms at the age of 7 months [18]. Therefore, we suggest that administration of NYT may be an effective therapeutic strategy to ameliorate physical frailty in the elderly.

## Conclusions

Ninjin'yoeito and its components have multifunctional and beneficial effects on several diseases. In this study, NYT administration was found to prevent atrophy of muscle cross-sectional area, alleviate inflammatory conditions caused by suppression of IL-1β and increase IL-1RA in the brain and skeletal muscle tissue, and reduce age-related fatigue-like conditions in mice. In addition, it also showed effects in preventing age-related weight loss, anorexia, and loss of motor function. The findings suggest that NYT administration may be a promising therapeutic strategy to improve physical frailty in the elderly.

## Supporting information

**S1 Checklist. Humane endpoints checklist.**
(PDF)

**S2 Checklist. The ARRIVE guidelines checklist.**
(PDF)

**S1 Fig. 3D-HPLC pattern of ninjin'yoeito (TJ-108, Tsumura & Co).**
(TIFF)

**S2 Fig. Average dose of NYT for the entire period.**
(TIFF)

**S3 Fig. Locations selected for quantification.**
(TIFF)

**S4 Fig. Changes in blood biomarkers.**
(TIFF)

**S1 File. Original underlying images for all Western blots.**
(DOCX)

**S1 Data. Raw date.**
(XLSX)

## Acknowledgments

The authors thank all members of the Tsumura Kampo Research Institute group for discussions. We would like to thank Editage (www.editage.jp) for English language editing.

## Author Contributions

**Conceptualization:** Shotaro Otsuka, Ryoma Matsuzaki, Ikuro Maruyama.

**Formal analysis:** Shotaro Otsuka, Ryoma Matsuzaki.

**Funding acquisition:** Keita Mizuno, Yosuke Matsubara.

**Investigation:** Shotaro Otsuka, Ryoma Matsuzaki, Shogo Kakimoto, Yuta Tachibe, Takuya Kawatani, Seiya Takada, Akira Tani, Kazuki Nakanishi, Teruki Matsuoka, Yuki Kato, Masaki Inadome, Nao Nojima.

**Methodology:** Shotaro Otsuka, Ryoma Matsuzaki, Ikuro Maruyama.

**Writing – original draft:** Shotaro Otsuka, Ryoma Matsuzaki.

**Writing – review & editing:** Harutoshi Sakakima, Ikuro Maruyama.

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
