## [Decision Letter · Decision Letter 0]

4 Mar 2024

PONE-D-24-03146Ninjin’yoeito reduces fatigue-like conditions by alleviating inflammation of the brain and skeletal muscles in aging micePLOS ONE

Dear Dr. Otsuka,

Thank you for submitting your manuscript to PLOS ONE. After careful consideration, we feel that it has merit but does not fully meet PLOS ONE’s publication criteria as it currently stands. Therefore, we invite you to submit a revised version of the manuscript that comprehensively addresses the points raised by both reviewers.

We look forward to receiving your revised manuscript.

Kind regards,

Andre van Wijnen

Academic Editor

PLOS ONE

“I have read the journal's policy and the authors of this manuscript have the following competing interests: KM and YM are company employees of the funding partner, Tsumura & Co. However, they have only funded the study and have no role in the study design, data collection and analysis, decision to publish, or preparation of the manuscript.”

In your cover letter, please note whether your blot/gel image data are in Supporting Information or posted at a public data repository, provide the repository URL if relevant, and provide specific details as to which raw blot/gel images, if any, are not available. Email us at plosone@plos.org if you have any questions."

5. We notice that your supplementary figures are uploaded with the file type 'Figure'. Please amend the file type to 'Supporting Information'. Please ensure that each Supporting Information file has a legend listed in the manuscript after the references list.

Reviewers' comments:

Reviewer's Responses to Questions

**Comments to the Author**

1. Is the manuscript technically sound, and do the data support the conclusions?

Reviewer #1: Partly

Reviewer #2: Yes

2. Has the statistical analysis been performed appropriately and rigorously? 

Reviewer #1: Yes

Reviewer #2: Yes

3. Have the authors made all data underlying the findings in their manuscript fully available?

Reviewer #1: Yes

Reviewer #2: Yes

4. Is the manuscript presented in an intelligible fashion and written in standard English?

Reviewer #1: Yes

Reviewer #2: Yes

5. Review Comments to the Author

Reviewer #1: The manuscript elaborated the effect of NYT on the fatigue through behavioral test and biomarker based on the inflammation status. In fact, it’s well known that the close relationship between the inflammation status and fatigue, so, it is suggested that the pathway for inflammation need to be explored. In addition, I found the marked red content, I believed that author had revised the manuscript, but there were some confusion.

1. SAMP8 is aggressive, 2 or 3 mice in one cage, if is there the mice fight even hurt? If the mice fight each other, it would increase the fatigue, and if hurt, it might increase the inflammation status. And, the author also found the intake NYT or foods for each mice were different.

2.the survival mice on the 7-month age should be explained

3. the dosage of NYT of 0.3×food, how to get this dosage

Reviewer #2: This paper is very interesting as it shows that ninjin’yoeito improves age-related inflammatory conditions in both the central and peripheral tissues and reduces fatigue. It has also been shown that SAMP8 can serve as a model animal for fatigue, making this an important paper that will contribute to the development of future research. In addition, as this study showed that a treatment method using ninjin'yoeito can improve age-related fatigue, it is expected that it will be applied to recovery from fatigue in the elderly.

I would like to ask some questions because there are some things that are unclear.

Materials and methods

Animals:

1. There is a statement that SAMP8 mice were purchased at 3 months old, but wouldn't the breeding environment have changed significantly?

2. Mice are kept in groups of 2-3 per cage, but wouldn't the meaning of the breeding environment be different if two mice are housed one-on-one versus three or more mice to form a society?

NTY preparation:

3. The authors divided the experiment groups into four. This is often a factor that confounds the interpretation of results. There are difficulties in using younger SAMR for comparison. The authors use it differently depending on the purpose, but except for Figure 4 the control is R1C. It is thought that R1N should also be included in the graph data.

4. There is a statement that it is mixed into the feed at a concentration of 3% (w/w), but is this the appropriate concentration? Are the density settings quoted from somewhere? Please indicate the basis for setting the concentration.

5. Is it correct to understand that SAMR is a mouse of normal age? Compared to SAMP8, when mice of the same age are used in experiments, SAMR, which is a younger mouse, seems to have a better appetite. If this is the case, it is thought that the dose of NTY may be different from that of SAMP8 since it is administered ad libitum. Does that mean that the problem was corrected by correcting the weight?

Quantiative analysis of immunostained sections, and S3 Figure B:

6. Does the region selected for analysis represent the center of the striatum? From S3 Figure A, the staining is unclear and the site is unclear.

The striatum also has site-specific roles. For example, the ventral striatum (VS) is involved in motivated behavior. If you chose the central part of the striatum, what was the reason? Also, why did they choose the striatum in the brain? Other issues related to exercise include neural circuits with the basal ganglia, motor control, and reduction in the amount of activity. Please discuss why you chose the striatum, along with the results of behavioral analysis.

7. Immunohistochemical staining was used to distinguish between fast and slow muscles. Was there any deviation in the distribution? For example, there are many slow-twitch muscles in the deep layers (close to bones), and on the contrary, there are many fast-twitch muscles in the superficial layers. Why did you choose the part you examined?

Results

p.19, L.325-327:

8. Is the peak weight of SAMR1 at 6 months of age, coinciding with the peak of SAMP8? Since SAMP8 accelerates aging, it may have reached its peak earlier. Is it appropriate to compare SAMP and SAMR at this month of age?

p.21, L.356-360:

9. Why did only corticosterone show high levels? What is the difference from other blood fatigue markers? It is stated in the discussion that it reflects the period immediately after exercise, but how long does corticosterone reflect fatigue? Or are these differences based on differences in the principles of fatigue? Please explain the difference.

p.23, L.397- p.24, L.406:

10. The results of administering NYT to SAMR and its control group are described here. If this result was shown first, it would be easier to understand that there is no RIN in the graph of Figure 1-3. I would like to suggest that the order of the structure of the paper be changed.

p.24, L405-406:

11. SAMR1 has no effect of NYT, while SAMP8 has been shown to have the effect. Is there a difference in fatigue between young and old people? Also, are there differences in the mechanism of fatigue recovery? Please answer about them.

Figure 3:

12. Although the areas of muscle fibers are compared, the difference in number is more obvious than the area, especially in the photo of “D” Type 1 fibers. Isn't it a difference in area due to a difference in number?

Also, am I correct in my understanding that P8N can maintain the area of fast-twitch muscles? Analysis of the expression of switch proteins such as PGC1α can reveal the slow muscle fibers associated with aging.

Figure 4:

13. Regarding “K”, fast twitch fibers generally replace slow twitch fibers as people age in the human. In mice treated with NYT, this did not occur and the number of fast-twitch muscles was maintained. Does it have an anti-aging effect on muscles?

Minor comments:

p.12, L.188：

I referred to literature [18], but there is no mention of the brightness of the room. Please indicate the measurement conditions.

P.19, L.317:

P<0.05: Shouldn't "P" be lowercase? p<0.05

p.25, L.427, 429:

Fig 5D, F, p<0.01, Fig 5D, F, p<0.05 Isn't it "E" instead of "F"? please confirm.

Materials and methods, Behavioral test:

Mice are nocturnal, so isn't it more reliable to perform behavioral tests at night?

Materials and methods, Histology and immunohistochemistry:

There is no description of the location of the coronal section of the brain. It is usually written as how many mm forward from bregma.

Figure 1:

The text compares SAMR and SAMP. However, the way graph "B" is drawn is not drawn as graph "D." Please check the graphs of other figures again. It is a horizontal bar that shows the p value.

Figure 4:

Regarding “A”, there was no difference between the three groups, is there any particular need to show it as a photograph? Also, if you want to show it, why not show the positive cells with an arrow head?

6. PLOS authors have the option to publish the peer review history of their article (what does this mean?). If published, this will include your full peer review and any attached files.

Reviewer #1: No

Reviewer #2: No

---

## [Author Response · Author response to Decision Letter 0]

3 Apr 2024

To the Editor.

The latest Competing Interests Statement was included in the cover letter.

Word files showing blot images and Excel files showing raw data have been uploaded as supporting information.

Supporting figures have been uploaded in the supporting information file.

 A file containing the changes has been uploaded to the 'Revised Manuscript with Track Changes' file. Clean copy manuscripts are marked in red where changes have been made.

To Reviewers

Thank you for peer review.

We have uploaded our response to your remarks in a 'Response to Reviewers' file.

---

## [Decision Letter · Decision Letter 1]

2 May 2024

Ninjin’yoeito reduces fatigue-like conditions by alleviating inflammation of the brain and skeletal muscles in aging mice

PONE-D-24-03146R1

Dear Dr. Otsuka,

We’re pleased to inform you that your manuscript has been judged scientifically suitable for publication and will be formally accepted for publication once it meets all outstanding technical requirements.

Kind regards,

Andre van Wijnen

Academic Editor

PLOS ONE

Additional Editor Comments (optional):

Editorial comments: the authors have adequately revised this paper.

Reviewers' comments:

Reviewer's Responses to Questions

**Comments to the Author**

1. If the authors have adequately addressed your comments raised in a previous round of review and you feel that this manuscript is now acceptable for publication, you may indicate that here to bypass the “Comments to the Author” section, enter your conflict of interest statement in the “Confidential to Editor” section, and submit your "Accept" recommendation.

Reviewer #2: (No Response)

2. Is the manuscript technically sound, and do the data support the conclusions?

Reviewer #2: Yes

3. Has the statistical analysis been performed appropriately and rigorously? 

Reviewer #2: (No Response)

4. Have the authors made all data underlying the findings in their manuscript fully available?

Reviewer #2: Yes

5. Is the manuscript presented in an intelligible fashion and written in standard English?

Reviewer #2: Yes

6. Review Comments to the Author

Reviewer #2: Before the paper is published, all authors should carefully check it again to ensure that there are no careless mistakes.

7. PLOS authors have the option to publish the peer review history of their article (what does this mean?). If published, this will include your full peer review and any attached files.

Reviewer #2: No

---

## [Editor Report · Acceptance letter]

8 May 2024

PONE-D-24-03146R1 

PLOS ONE

Dear Dr. Otsuka, 

I'm pleased to inform you that your manuscript has been deemed suitable for publication in PLOS ONE. Congratulations! Your manuscript is now being handed over to our production team.

Kind regards, 

on behalf of

Dr. Andre van Wijnen 

Academic Editor

PLOS ONE